# Dynamics of the viral community on the surface of a French smear-ripened cheese during maturation and persistence across production years

Thomas Paillet,[1] Quentin Lamy-Besnier,[2] Clarisse Figueroa,[1] Marie-Agnès Petit,[2] Eric Dugat-Bony[1]

**ABSTRACT** The surface of smear-ripened cheeses constitutes a dynamic microbial ecosystem resulting from the successive development of different microbial groups such as lactic acid bacteria, fungi, and ripening bacteria. Recent studies indicate that a viral community, mainly composed of bacteriophages, also represents a common and substantial part of the cheese microbiome. However, the composition of this community, its temporal variations, and associations between bacteriophages and their hosts remain poorly characterized. Here, we studied a French smear-ripened cheese by both viral metagenomics and 16S metabarcoding approaches to assess both the succession of phages and bacterial communities on the cheese surface during cheese ripening and their temporal variations in ready-to-eat cheeses over the years of production. We observed a clear transition of the phage community structure during ripening with a decreased relative abundance of viral species (vOTUs) associated with *Lactococcus* phages, which were replaced by vOTUs associated with phages infecting ripening bacteria such as *Brevibacterium, Glutamicibacter, Pseudoalteromonas,* and *Vibrio.* The dynamics of the phage community was strongly associated with bacterial successions observed on the cheese surface. Finally, while some variations in the distribution of phages were observed in ready-to-eat cheeses produced at different dates spanning more than 4 years of production, the most abundant phages were detected throughout. This result revealed the long-term persistence of the dominant phages in the cheese production environment. Together, these findings offer novel perspectives on the ecology of bacteriophages in smear-ripened cheese and emphasize the significance of incorporating bacteriophages in the microbial ecology studies of fermented foods.

**IMPORTANCE** The succession of diverse microbial populations is critical for ensuring the production of high-quality cheese. We observed a temporal succession of phages on the surface of a smear-ripened cheese, with new phage communities showing up when ripening bacteria start covering this surface. Interestingly, the final phage community of this cheese is also consistent over large periods of time, as the same bacteriophages were found in cheese products from the same manufacturer made over 4 years. This research highlights the importance of considering these bacteriophages when studying the microbial life of fermented foods like cheese.

**KEYWORDS** bacteriophages, cheese, microbial ecology, fermented food, metavirome, metagenomics, succession, persistence

Smear-ripened cheeses develop a typical viscous, red-orange smear on their surface during ripening (1). In France, 14 out of the 46 protected designation of origin (PDO) cheeses belong to this category, that is, Abondance, Beaufort, Chevrotin, Comté, Epoisses, Langres, Livarot, Maroilles, Mont d'Or, Morbier, Munster, Ossau-Iraty, Pont

Address correspondence to Eric Dugat-Bony, eric.dugat-bony@inrae.fr.

The authors declare no conflict of interest.

l'Evêque, and Reblochon. These cheeses collectively accounted for 61% of the total annual volume of French PDO cheeses marketed, with 128 kilotons in 2021 (https://www.inao.gouv.fr/eng/Publications/Donnees-et-cartes/Informations-economiques). Due to their unique ripening process, involving frequent washes with saline and/or alcoholic solutions, smear-ripened cheeses host a distinct microbiota compared to other cheeses (1). This microbiota is composed of lactic acid bacteria (LAB, mainly starter cultures added at the beginning of the process for milk acidification), yeasts, and salt-tolerant bacteria belonging to the Actinomycetota, Bacillota, and Pseudomonadota phyla. It is considered to be responsible for the typical flavor and organoleptic properties of this type of cheese (2). Over the past two decades, numerous studies have been conducted to describe the composition of this microbiota using culture-dependent methods (3, 4), molecular fingerprinting (5–8), and more recently amplicon-based metagenomics, also commonly referred to as metabarcoding (9–12).

Time series studies also revealed the microbial succession that occurs on the surface of smear-ripened cheeses during the maturation process (8, 13, 14). Lactic acid bacteria (LAB), usually originating from starter cultures, grow first in the milk and represent the dominant microorganisms in the curd. Yeasts, for example, *Debaryomyces hansenii* and *Geotrichum candidum*, which exhibit acid tolerance and metabolize lactate, subsequently colonize the cheese surface, resulting in its deacidification. With the pH increase, the establishment of a diverse bacterial community originating from raw milk, ripening cultures, or the manufacturing environment, is progressively observed. The most common bacterial taxa detected at the end of ripening on smear-ripened cheese belong to coryneform bacteria (e.g., species of the *Glutamicibacter*, *Brevibacterium*, *Corynebacterium,* or *Brachybacterium* genera), *Staphylococcus* species and halophilic or halotolerant gram-negative bacteria (e.g., species of the *Psychrobacter*, *Halomonas*, *Pseudoalteromonas, Hafnia, Vibrio*, *Pseudomonas,* or *Proteus* genera) (15).

Microbial successions in cheese during the ripening process result from the interplay between abiotic factors, microbial interactions, and complex ecological processes such as dispersion, diversification, environmental selection, and ecological drift (16). While abiotic factors' impact on cheese microbial community assembly has been extensively studied (17, 18), and some important microbial interactions have been revealed as well (19–21), the effects of bacteriophages (or phages), that is, viruses that infect and replicate within bacteria, remain largely understudied. Their role in community assembly has been highlighted in many natural ecosystems (22) and recent work suggests they may affect microbial community dynamics in fermented foods (23, 24), especially through the elimination of key bacterial populations.

In the dairy industry, the impact of LAB phages is well documented because their lytic activity can disturb the milk acidification step, causing delays in production and even total loss of production (25, 26). Consequently, the most studied dairy phages are *Lactococcus*, *Streptococcus*, *Lactobacillus,* and *Leuconostoc* phages, infecting the main starter cultures (27). Viral metagenomics has been recently employed to characterize the bacteriophage communities in dairy samples, including whey (28) and cheeses (29, 30). These investigations have demonstrated that viral communities are not restricted to LAB phages, but encompass a diverse array of phages that could infect non-inoculated and ripening bacteria during cheese production. The recent application of shotgun metagenomics combined with metaHiC on three washed-rind cheeses also enabled to associate hundreds of putative phage contigs with metagenome-assembled genomes (MAGs) of their bacterial hosts, particularly those belonging to Actinomycetota and Pseudomonadota phyla (14).

Beyond metagenomics, only a few studies have documented the isolation of such virulent phages infecting ripening bacteria in cheese, that is, targeting *Propionibacterium freudeunreichii* and *Brevibacterium aurantiacum* (31–33). We recently isolated five additional virulent phages, targeting *Glutamicibacter arilaitensis* (phages Montesquieu and Voltaire), *Brevibacterium aurantiacum* (phage Rousseau), *Psychrobacter aquimaris* (phage d'Alembert), and *Leuconostoc falkenbergense* (phage Diderot), from the surface of

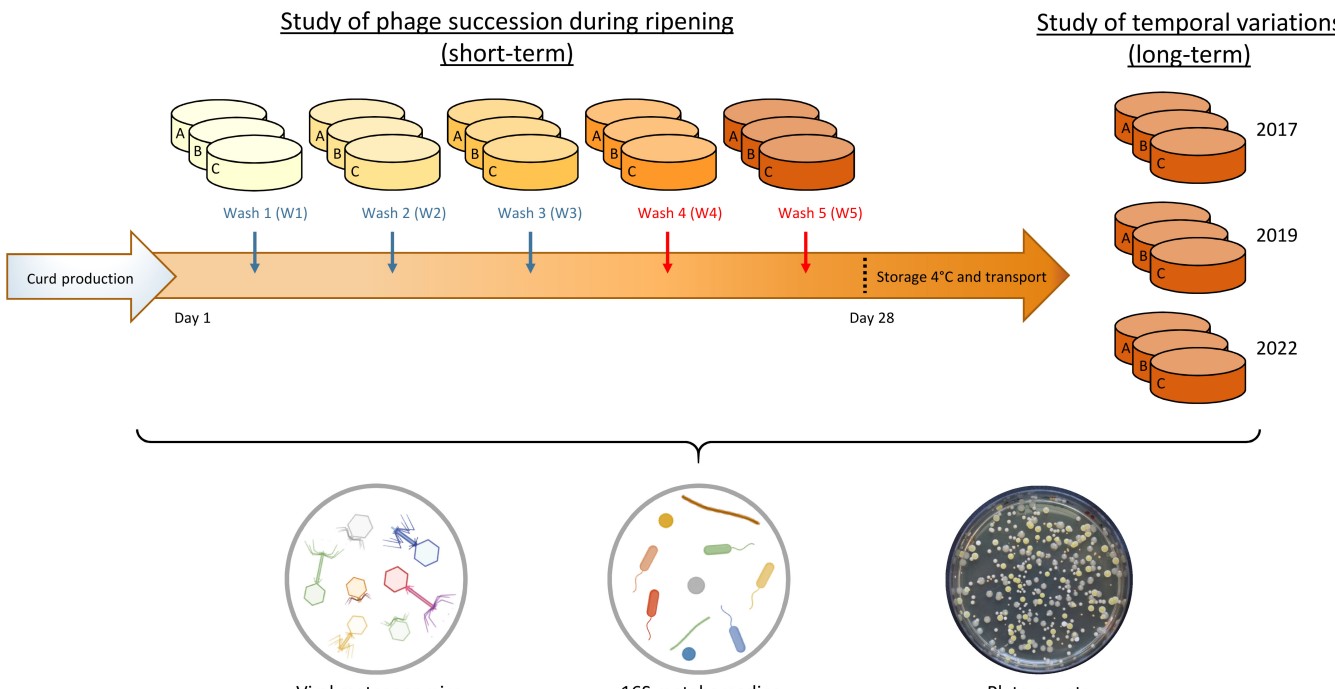

**FIG 1** Schematic representation of the experimental plan followed for the study of temporal variation in the cheese virome composition. For the short-term study, cheese samples were collected in triplicate after each washing operation during the ripening process (blue arrows indicate washes performed with a NaCl solution, while red arrows indicate washes performed with a NaCl solution containing alcoholic liquor). For the long-term study, cheese samples were collected in triplicate after storage and transport at three different production years. All samples were analyzed through viral metagenomics, 16S metabarcoding, and microbial plate counts.

a French smear-ripened cheese suggesting that viral infection is likely to occur on such ecosystems for most of the dominant bacteria (34). However, little is known about their ecology. Specifically, their temporal variations within the cheese and their relationships with bacterial successions remain largely unexplored while being essential knowledge to assess the overall impact of phage infections on community assembly.

In this study, we, therefore, extended our investigations on the same smear-ripened cheese to enhance our understanding of the temporal distribution of bacteriophages and their bacterial hosts on the cheese surface across two distinct timescales (Fig. 1). First, cheeses were obtained directly from the production facility at five ripening stages over 28 days (three biological replicates per stage), to analyze phage succession throughout the ripening process (short-term study). Second, ready-to-eat cheeses, that is, after storage and transport to the retail shop, of the same brand and variety were sampled in 2017, 2019, and 2022 (three biological replicates per production year), to assess the temporal variation of the phage community in this ecosystem over the years of production (long-term study). Our results reveal a shift in bacteriophage communities during cheese ripening, correlating with bacterial successions, and highlight the long-term persistence of dominant phages in the studied cheese production system.

## RESULTS

### Overview of the cheese surface virome data set

The viral fraction was extracted from the surface of 24 smear-ripened cheese samples (Fig. 1). Fifteen samples were obtained during the ripening process to investigate the ecological succession of phages (short-term study). Nine additional samples were collected post-transport and storage to examine the temporal variation in the phage community across production years (long-term study). Subsequently, the extracted viral fraction was subjected to viral metagenomics techniques.

The cheese viromes had an average sequencing depth of 11,585,587 sequencing reads and were assembled into 5,753 contigs. Viral contigs were selected by considering 2 kb as the minimum size for viral sequence fragments and using a combination of viral sequence detection tools (see Materials and Methods). After clustering based on sequence homology, 331 species-level viral operational taxonomic units (vOTUs) were obtained (Table 1). The vast majority of the vOTUs (i.e., 284) were detected in samples from both the short-term and long-term studies. The percentage of reads mapping to these vOTUs reached 91.8% on average, revealing the high quality of virome assemblies. Detailed information about the vOTUs including their size and annotations is available in Table S1. The vOTUs sizes ranged from 2,002 to 144,530 bp, 79 contigs (24%) being >10 kb (Fig. 2A). This included 25 vOTUs (7.6%) flagged as complete genomes and 12 (3.6%) as high-quality genomes according to CheckV (35). In all, 27 vOTUs (8.2%) were identified as temperate phages by Vibrant (36) if they were excised from a larger contig or if they encode an integrase. A bacterial host was successfully predicted at the genus level for most vOTUs (272/331, i.e., 82.2%), spanning three bacterial phyla, namely Actinomycetota, Bacillota, and Pseudomonadota (Fig. 2B). *Lactococcus* was the predicted genus containing the highest phage diversity (182 vOTUs), followed by *Leuconostoc* (23 vOTUs). *Pseudoalteromonas* (18 vOTUs), *Psychrobacter* (17 vOTUs), and *Brevibacterium* (6 vOTUs). Only 40.4% of vOTUs (134/331) shared sequence homology (>30% identity ×coverage based on BLAT score) with a known dairy phage (Table S2). However, those vOTUs represented 97% of the relative abundance observed in the data set (Table 2) showing that most of the dominant phages present in this ecosystem have isolated and characterized relatives. Among them were present *Lactococcus* phages from the 949 (*Audreyjarvisvirus* genus), 936 (*Skunavirus* genus), and P335 groups, as well as the five virulent phages isolated from the same type of cheese (same variety, same brand) in our previous work (34), namely *Glutamicibacter* phages Montesquieu and Voltaire, *Brevibacterium* phage Rousseau, *Psychrobacter* phage D'Alembert, and *Leuconostoc* phage Diderot (*Limdunavirus* genus).

## Successional dynamics of the cheese virome throughout the ripening process

After phage identification, our next objective was to follow the phage communities throughout the ripening process, with a particular focus on the washing steps (short-term study). In the manufacturing process of the studied cheese, the first three washing operations (named W1, W2, and W3) utilize a NaCl solution, while the last two operations (W4 and W5) utilize a NaCl solution containing increasing concentrations of alcoholic liquor (Fig. 1). The data set consisted in 15 samples (triplicate samples for each washing operation) and 318 vOTUs (Table 1).

The viral diversity was estimated by the Shannon index, which varied from 2.631 to 3.442 across samples (median = 2.969). A slight but non-significant ($P > 0.05$, Kruskal-Wallis test; Fig. 3A) decrease in the Shannon index was noted from W3 onwards, suggesting there was no major alteration of the viral diversity during ripening. An overall sparsity of 29% (percentage of 0 values in the matrix) was observed on the abundance table reflecting that all the studied cheese viromes shared most of the detected vOTUs. However, structural variation in the virome composition was assessed by computing Bray-Curtis dissimilarity (Fig. 3B) and the PERMANOVA test indicated a significant effect of the washing operation ($P = 0.001$, $r^2 = 0.658$).

**TABLE 1** Summary statistics about the metavirome data set

| | Complete data set (24 samples) | Short-term study (15 samples) | Long-term study (9 samples) |
|---|---|---|---|
| Total number of vOTUs > 2 kb | 331 | 318 | 297 |
| Number of vOTUs with relative abundance >0.005% | 157 | 168 | 99 |
| Percentage of reads mapping against vOTUs (±SD) | 91.8 ± 12.1 | 94.3 ± 7.0 | 87.6 ± 17.4 |

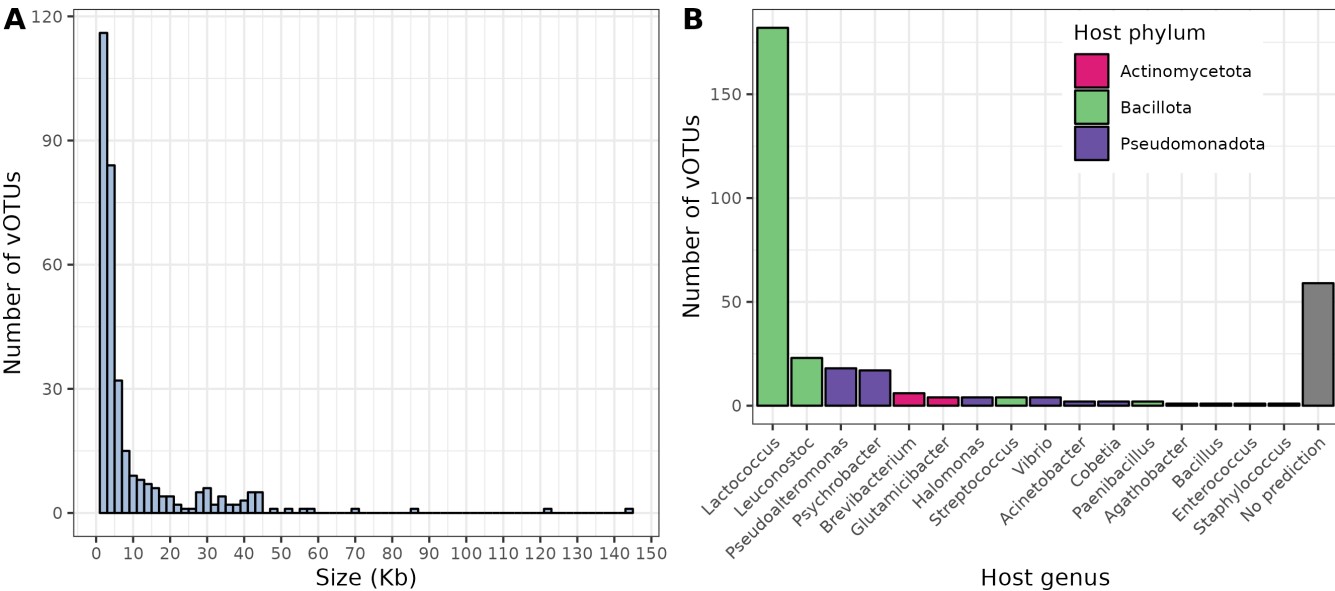

**FIG 2** Characteristics of the cheese virome data set. (A) Size distribution of the vOTUs. (B) Distribution of the vOTUs per bacterial host genus.

To explore further this change, we selected vOTUs with a normalized relative abundance above 0.005% on average (168 contigs) and represented their distribution across samples in a heatmap (Fig. 3C). A clear change in the structure of the vOTUs distribution appeared over time. Based on the total within the sum of the square (Fig. S1), we delineated five vOTUs clusters (labeled I to V in Fig. 3C). Clusters I and II were characterized by vOTUs whose abundance did not vary with the ripening and that corresponded mainly to virulent *Lactococcus* phages belonging to the 936 group (*Skunavirus* genus). Cluster V contained vOTUs whose abundance decreased with ripening and that corresponded to other *Lactococcus* phages, close to the P335 group containing both temperate and virulent members. Finally, clusters III and IV were essentially composed of a few vOTUs that increased in relative abundance with ripening. These mainly corresponded to phages targeting ripening bacteria such as *Brevibacterium*, *Glutamicibacter*, *Pseudoalteromonas,* and *Vibrio*, and non-starter lactic acid bacteria (NSLAB) such as *Leuconostoc*. These few vOTUs corresponded to uncharacterized

**TABLE 2** Most abundant dairy phages detected on the cheese metavirome[a]

| Closest relatives | Relative abundance[b] |
| --- | --- |
| *Lactococcus* 949 group (*Audreyjarvisvirus*) | 0.56 |
| **Glutamicibacter Montesquieu** | **0.23** |
| *Lactococcus* 936 group (*Skunavirus*) | 0.18 |
| *Lactococcus* P335 group | 1.54E−03 |
| **Brevibacterium Rousseau** | **1.08E−03** |
| **Psychrobacter D'Alembert** | **1.74E−04** |
| **Leuconostoc Diderot (Limdunavirus)** | **1.14E−04** |
| **Glutamicibacter Voltaire** | **1.70E−05** |
| *Streptococcus* 987 group | 7.87E−06 |
| *Lactococcus* KSY1 group (*Chopinvirus*) | 5.94E−07 |
| Non-dairy phages[c] | 0.03 |

[a]Phages that have been previously isolated from the same cheese variety are highlighted in bold. When classification is available at the International Committee on Taxonomy of Viruses (ICTV), the viral genus is denoted in parenthesis.
[b]The relative abundance value used here is not normalized by genome size, it corresponds to the relative proportion of reads mapping to all vOTUs belonging to a given phage group present in our in-house dairy phage database (Table S2).
[c]vOTUs without any match to our in-house dairy phage database (30% identity × coverage minimal cutoff).

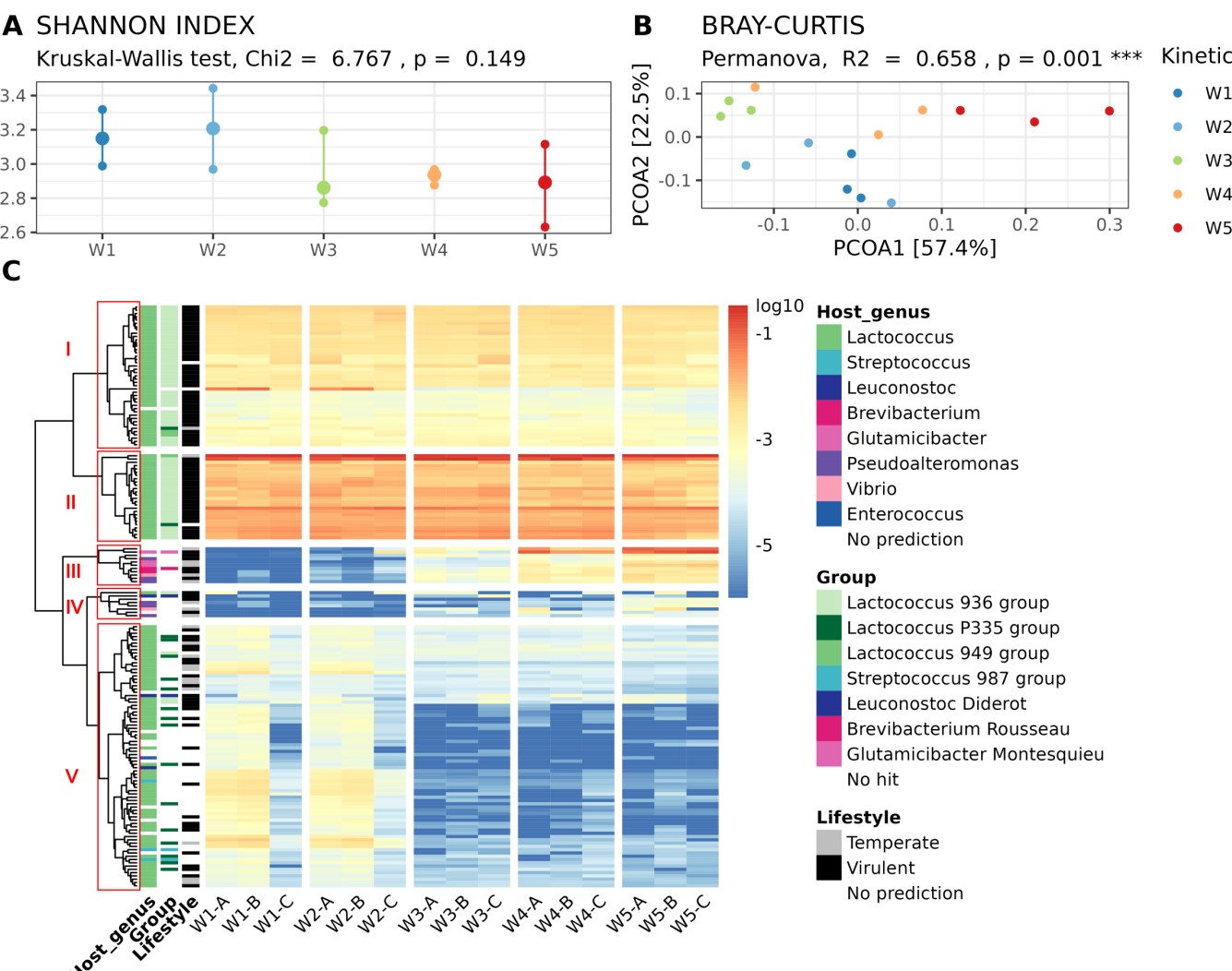

**FIG 3** Viral succession on the cheese surface throughout the ripening process. (A) Comparison of the viral diversity, estimated by the Shannon index, at the different ripening steps. (B) Principal coordinate analysis of the Bray-Curtis dissimilarity. Samples are colored according to the washing operation. (C) Heatmap representing the normalized relative abundance of the most abundant vOTUs in the different samples. The colors on the heatmap represent the log-transformed relative abundance and range from blue (low relative abundance) to red (high relative abundance). Host genus, phage group, and lifestyle of the different vOTUs are indicated when available. Hierarchical clustering using the complete linkage method was used to define the order of rows (vOTUs). The five vOTUs clusters detected in the data set are boxed in red and labelled I to V.

phages, except *Glutamicibacter* phage Montesquieu, *Brevibacterium* phage Rousseau, and *Leuconostoc* phage Diderot which we previously isolated from the same type of cheese (34).

Both the principal coordinate analysis of the Bray-Curtis dissimilarity (Fig. 3B) and the heatmap visualization (Fig. 3C) revealed a comparable virome composition between samples collected from W1 and W2, as well as between those from W4 and W5. This observation suggests the succession of two predominant viral communities on the cheese surface during the ripening process, corresponding to these respective stages. Samples from W3 displayed a unique and typically intermediate composition. In these samples, clusters III and IV were found in higher relative abundance compared to W1-W2 but not as abundant as in W4-W5. Cluster V was also detected in lower abundance than in W1-W2 but at a level similar to W4-W5.

We finally applied the DESeq2 method (37, 38) to statistically identify which vOTUs differentiate the two predominant viral communities represented by W1-2

and W4-5 samples (Fig. S2). In total, 22 vOTUs were differentially abundant between the two communities (adjusted $P < 0.05$, log2 fold change $> 3$ or $< -3$). Interestingly, two groups of vOTUs emerged with readable outcomes: vOTUs of the phages infecting starter cultures, such as *Lactococcus* and *Streptococcus* phages, had a negative log2 fold change, meaning their relative abundances significantly decreased during the ripening process. Conversely, vOTUs of virulent phages targeting NSLAB and ripening bacteria (e.g., *Brevibacterium, Glutamicibacter, Pseudoalteromonas,* and *Vibrio*) as well as unknown phages were significantly more abundant in W4-5 samples compared to W1-2 samples (positive log2 fold change), reflecting their higher population level in the later ripening stages. Among them were two vOTUs with high similarity to *Glutamicibacter* phage Montesquieu (2017-3_NODE2; 47,703 bp; 100% identity × coverage by BLAT), and *Brevibacterium* phage Rousseau (2022-5_NODE4; 41,077 bp; 54% identity × coverage by BLAT but 98% identity at the nucleotidic level over a portion of >9 kb). In this group, we also detected a vOTU partially related to *Brevibacterium* phage AGM1 (SA2-0_NODE10; 37,148 bp; 3.5% identity × coverage by BLAT but 88% identity at the nucleotidic level over a portion of 1324 nt), which was isolated from a Canadian washed-rind cheese (32).

## Relationships between the cheese virome and bacteriome throughout the ripening process

Having observed a succession of viral communities on the cheese surface during the ripening process, we sought to investigate how it was associated with bacterial community changes. We therefore profiled bacterial communities on the same samples using 16S metabarcoding. A total of 773,059 reads were obtained and clustered into 195 operational taxonomic units (OTUs) assigned to 14 different bacterial genera (Fig. 4A). As observed for viruses, the composition of the bacteriome of the cheese surface varied through the ripening process and a clear transition in the bacterial community structure was detected from W3 onwards. *Lactococcus* was the dominant genus in samples from W1 and W2 and was progressively overgrown by typical surface aerobic bacteria such as members of *Psychrobacter*, *Vibrio*, *Glutamicibacter* and *Pseudoalteromonas* genera.

As metabarcoding data provide only relative proportions, bacterial counts were performed to introduce a quantitative aspect to the bacterial dynamics (Fig. 4B and C). We observed an approximately 2-log increase in aerobic bacteria between W1 and W5, rising from ~$10^8$ to ~$10^{10}$ CFU/g. By contrast, lactic acid bacteria remained stable over time (~$10^8$ CFU/g) throughout the entire experiment. We concluded that the observed change in the relative abundance of *Lactococcus* sequences through metabarcoding was primarily attributed to the growth of aerobic bacteria.

A Mantel test demonstrated the statistical relationships between the virome and bacteriome data sets ($P = 0.001$; $r = 0.3151$; 999 permutations). This indicates that changes observed on the phage community structure are associated with bacterial successions on the cheese surface. We further calculated Spearman correlations between the relative abundances of the main phage groups identified in the metavirome and of bacterial genera in the bacteriome to identify co-abundance or mutual-exclusion patterns (Fig. 4D). We observed several positive correlations between phages and their bacterial host genus. For example, *Psychrobacter* phage d'Alembert and *Glutamicibacter* phage Montesquieu exhibited positive correlations with several ripening bacteria including *Psychrobacter* and *Glutamicibacter* genera. *Brevibacterium* phage Rousseau correlated also with the same bacterial genera but not with *Brevibacterium,* likely due to its low detection level in the studied cheese. LAB phages belonging to KSY1, P335, and 987 groups, which are infecting *Lactococcus lactis* and/or *Streptococcus thermophilus*, were positively correlated with the *Lactococcus* genus.

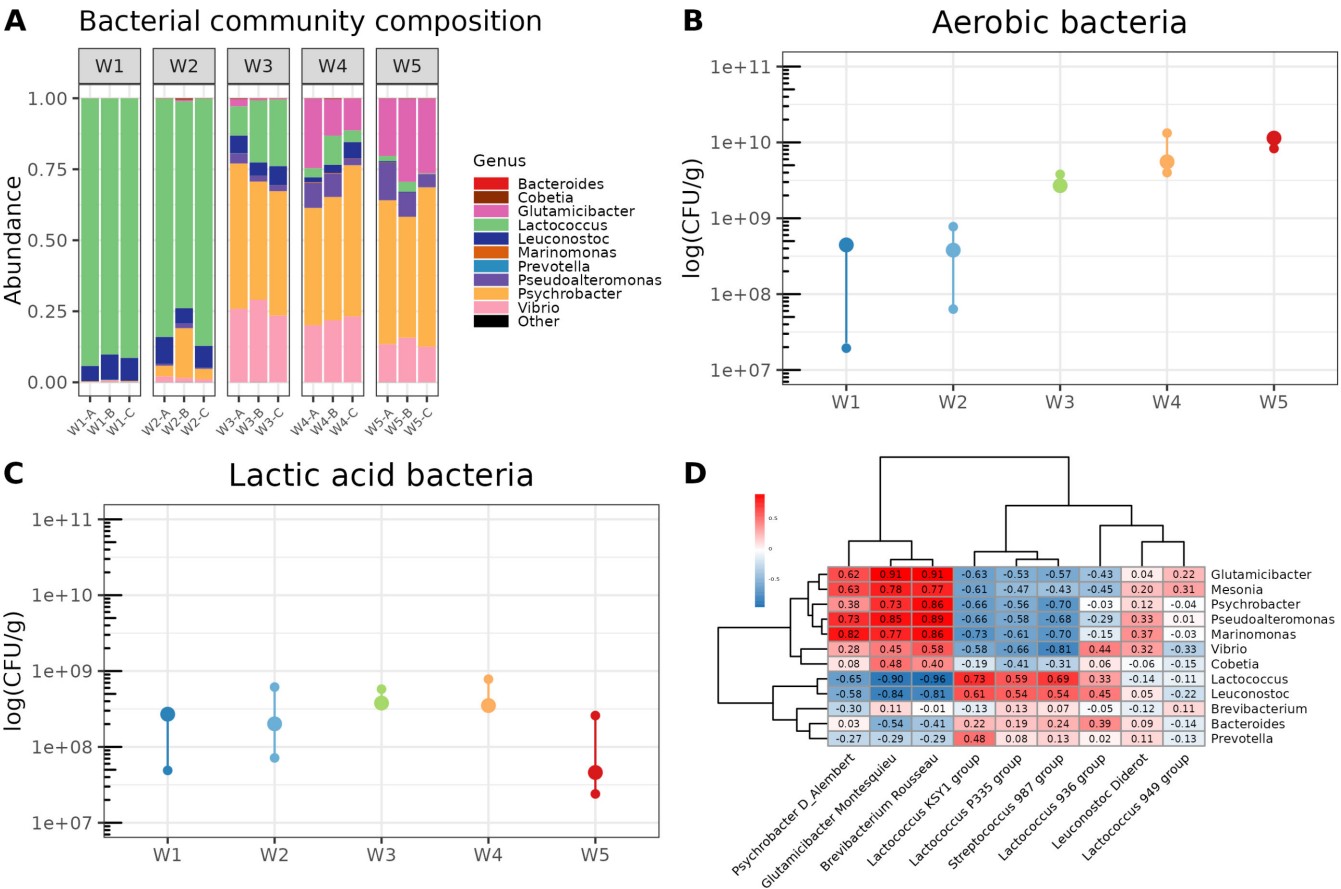

**FIG 4** Relationship between phage succession and the composition of cheese bacterial community during the ripening process. (A) The composition of the bacterial community was assessed by a metabarcoding approach targeting the V3-V4 regions of the 16S rRNA gene. Three different cheeses from the same batch were analyzed at each sampling point. Data were aggregated at the genus level. (B) Aerobic bacteria counts expressed in log(CFU/g). (C) Lactic acid bacteria counts expressed in log(CFU/g). (D) Heatmap of Spearman correlations (*r* values, ranging from −1 to +1) describing the relationship between the main phage groups detected in the virome data set and bacterial genus-level relative abundances. Hierarchical clustering using the complete linkage method was used to define the order of rows (bacterial genera) and columns (phage groups).

## Temporal variation of the cheese virome across production years

As phage succession was evidenced in the short term, specifically during the ripening process, we next wanted to test whether the composition of the virome remained comparable over multiple years of cheese production. To address this, we analyzed the long-term data set comprising 297 vOTUs detected on nine cheese samples of the same type and brand as the ones used for the short-term study (Table 1). These samples were collected after ripening, transport, and storage across three different production years (Fig. 1). We first focused on determining the proportion of vOTUs shared across production years. Utilizing the full data set without relative abundance filtering, we observed that 56.6% of the vOTUs (168) were consistently detected in all production years, with 75.4% (224) observed in at least two different production years (Fig. 5A). Notably, samples from the year 2022 had a higher number of unique vOTUs (48, 32.1% of the total) compared to samples from the years 2017 and 2019 (3 and 11, respectively). We also conducted the same analysis focusing on the most abundant vOTUs, defined by an average normalized relative abundance >0.005% (99 vOTUs in total) (Table 1; Fig. 5B). A large majority of the vOTUs, that is, 89.9%, were shared across all three production years and this value rose to 97% when considering only two different years. This underscores the stability of the cheese surface virome in terms of composition

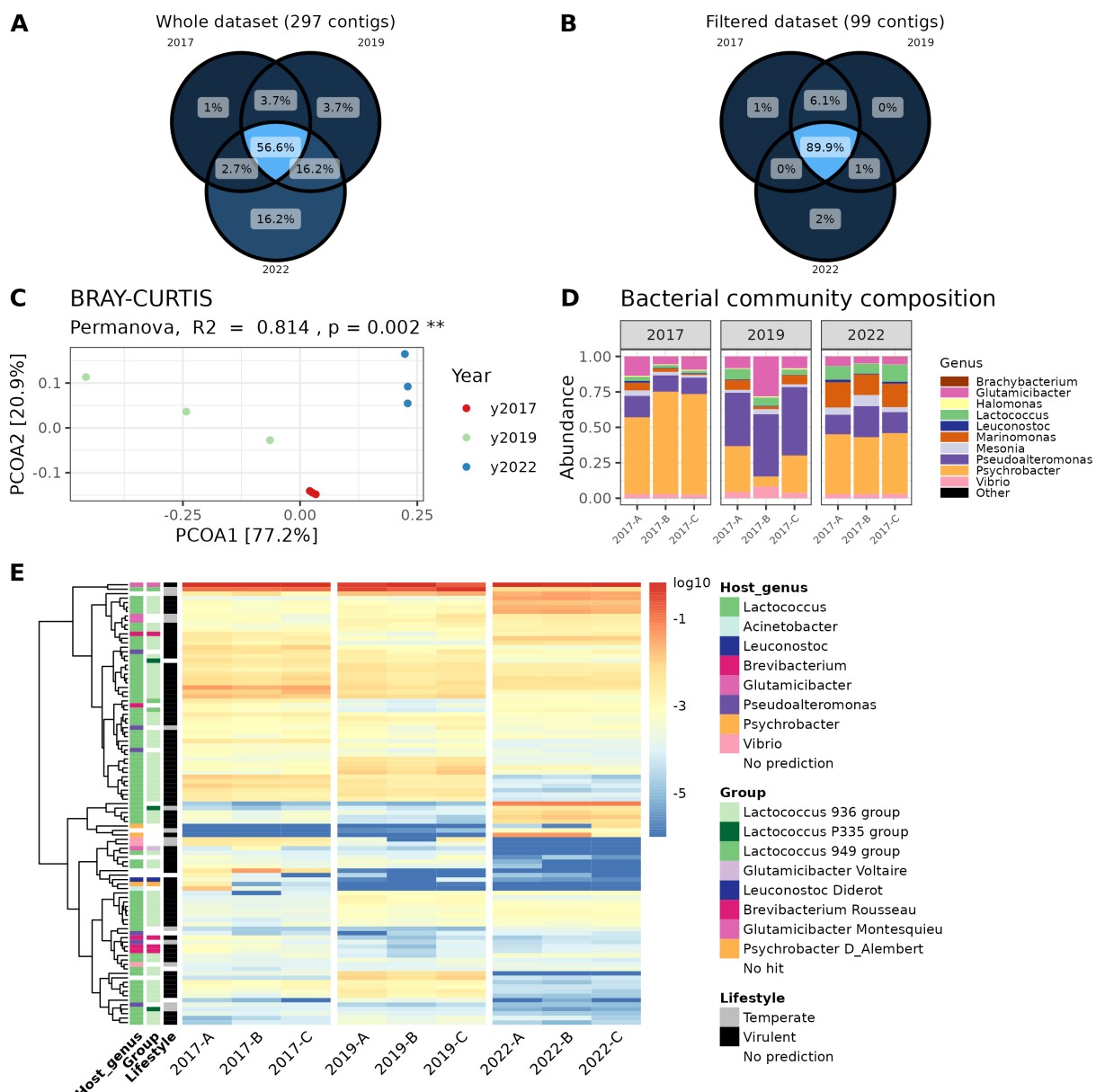

**FIG 5** Temporal variation of the cheese surface virome across production years. (A) Venn diagram constructed from the data set without filtering, comprising 297 vOTUs. (B) Venn diagram constructed from the filtered data set comprising 99 vOTUs with an averaged normalized relative abundance value >0.005%. (C) Principal coordinate analysis of the Bray-Curtis dissimilarity. Samples are colored according to the production year (2017, 2019, and 2022). (D) Composition of the bacterial community assessed by a metabarcoding approach targeting the V3-V4 regions of the 16S rRNA gene. Three different cheeses from the same production year were analyzed. Data were aggregated at the genus level. (E) Heatmap representing the normalized relative abundance of the most abundant vOTUs in each sample. The colors on the heatmap represent the log-transformed relative abundance and range from blue (low relative abundance) to red (high relative abundance). Host genus, phage group, and lifestyle of the different vOTUs are indicated when available. Hierarchical clustering using the complete linkage method was used to define the order of rows (vOTUs).

(presence/absence) and emphasizes the long-term persistence of dominant phages in the studied ecosystem.

The effect of the production year (2017, 2019, and 2022) on the structure of the cheese virome was further investigated by computing Bray-Curtis dissimilarity on the vOTU abundance table without relative abundance filtering (Fig. 5C). The principal coordinate analysis revealed a distinct grouping of samples consistent with the production years. The first axis discriminated samples from the year 2019 and samples

from the years 2017 and 2022. The second axis discriminated samples from the years 2017 and 2022. The PERMANOVA test confirmed a significant effect of the production year ($P = 0.002$, $R^2 = 0.814$) on the distance metrics indicating structural variations of the phage community across production years. Given the consistent presence of the most abundant vOTUs across all production years (Fig. 5B), such variations could be attributed to the fluctuations in the presence/absence of phages with lower abundance, as well as to the distribution of dominant phages in terms of relative abundances.

Heatmap visualization was then used to identify phages explaining such structural variations (Fig. 5E) in combination with differential abundance analysis (Fig. S3). Whereas the general profiles of all the viromes were very similar, a few vOTUs were more specifically detected in one or two production years. For example, the second most abundant vOTU in 2017 and 2019 (SS2-4_NODE1) which is highly similar to *Lactococcus* phage 949 (*Audreyjarvisvirus* genus) was statistically less abundant in 2022 (adjusted *P* < 0.05, log2 fold change < −3) contrary to a group of other distinct vOTUs also predicted to infect *Lactococcus* but either not identified, or assigned to the 936 (*Skunavirus* genus) and P335 groups. Regarding *Psychrobacter* phages, the vOTU closely related to phage d'Alembert (2017-0_NODE1) was more abundant in 2017 than in the two other years (adjusted *P* < 0.05, log2 fold change > 3). However, two vOTUs without sequence homology with known dairy phages, but predicted to infect also *Psychrobacter* (2022-0_NODE3 and 2022-5_NODE3), were detected later in 2022 in higher abundance (adjusted *P* < 0.05, log2 fold change > 3). Notably, the *Glutamicibacter* phage Voltaire was present in low abundance in 2017 and 2019 (0.036% and 0.004% relative abundance on average, respectively), but was not detected in 2022.

We also performed 16S metabarcoding analysis to determine whether compositional changes were observed in the bacterial community as well. The surface of the studied smear-ripened cheese was characterized by 10 dominant bacterial genera. Similar to what we observed for phages, these genera were consistently detected in all samples (Fig. 5D). However, variations in their relative abundance were detected across the three production years.

Together, our results indicate the presence of a core microbiome consisting of dominant microorganisms and their associated viruses that define the surface of the cheese under study. The composition of this core microbiome seems to persist in the long-term timescale, although changes in the microbial balance can be observed between production batches.

## DISCUSSION

Recent studies of cheese samples using viral metagenomics (29, 30) or exploration of cheese microbial metagenomes (14, 39) have revealed that the cheese environment harbors a diverse bacteriophage community whose targets go beyond lactic acid starter cultures. The previous isolation of a few representatives of these non-starter phages (32–34) suggests that phage infection occurs in this ecosystem during the ripening process. Recently, the deliberate addition of one of those phages, *Brevibacterium* phage AGM9, was proven to slow down the development of the orange rind color in a model system mimicking a smear-ripened cheese (40). In the present study, we described the temporal variation of the viral community of French smear-ripened cheese at two timescales: short term (28 days corresponding to the duration of the ripening process) and long term (spanning 4 years of production). The virome was predominantly composed of vOTUs associated with a variety of *Lactococcus* phages as well as the *Glutamicibacter* phage Montesquieu, a virulent phage we had previously isolated from the same cheese variety (34). This phage targets the ripening bacterium *Glutamicibacter arilaitensis*.

Several other known dairy phages, such as *Brevibacterium* phage Rousseau, *Leuconostoc* phage Diderot, *Psychrobacter* phage d'Alembert, *Glutamicibacter* phage Voltaire, and a novel vOTU displaying minimal sequence homology to *Brevibacterium* phage AGM1 were also detected, but at lower abundance. Notably, while known dairy phages represented 97% of the relative abundance in our data set, 197 vOTUs (59.6%

of the total) remained unidentified and 49 (17.8%) had no predicted host. The diversity of these low-abundant phages is therefore not yet completely sampled and, in particular, our results underscore the necessity to isolate a more comprehensive collection of phages from cheese and other fermented foods in general. For example, a few vOTUs partially aligned with genomic sequences from phages infecting halotolerant bacteria but isolated from natural environments (e.g., *Pseudoalteromonas*, *Halomonas*, *Vibrio*, and *Proteus* species). Even though these bacteria are not intentionally inoculated into smear-ripened cheese, they are frequently detected in such products and tend to dominate by the end of the ripening process (15). Their phages should therefore be looked for in future isolation initiatives.

The dynamics of bacterial and fungal communities have been extensively studied in a wide diversity of cheese products during ripening (13, 41–45), enabling us to accurately describe the successions of cellular organisms occurring in cheese production processes (16). In this study, we present the first analysis of viral dynamics throughout cheese ripening. Similar to the observed transitions in bacteria and fungi, there was a distinct shift in viral composition throughout the ripening process. Some vOTUs associated with LAB starter phages, primarily targeting *Lactococcus lactis*, were progressively replaced by vOTUs specific to phages that infect ripening bacteria. We observed a strong correlation between the dynamics of phage and bacterial communities during the ripening process. Specifically, the relative abundance of several phages, such as *Glutamicibacter* phage Montesquieu, *Psychrobacter* phage d'Alembert, and *Lactococcus* phages belonging to the 936 (*Skunavirus* genus), KSY1, and P335 groups was positively correlated to the relative abundance of their predicted bacterial hosts. This observed positive correlation contrasts with the anticipated negative correlation that might have been expected if virulent phages were responsible for eliminating key members of the bacterial community during cheese ripening. This result therefore raises questions about the overall impact of phages on bacterial successions in cheese during ripening. However, it is important to note that phage susceptibility is generally strain dependent, as demonstrated in previous studies on LAB strains (summarized in reference 46) and in undefined starter cultures where multiple strains of the same species stably coexist with distinct phages (47–49). The narrow host range observed for phages infecting ripening bacteria recently isolated from cheese rind also corroborates this result (32, 34). In the present work, the composition of the bacterial community was assessed at the genus level, thus preventing capturing such strain-level diversity. Future investigations at the strain level would therefore be highly relevant to better understanding the ecological role of phages in cheese rind microbial communities.

Surprisingly, despite the changes in bacterial and viral communities' composition during the production process, the relative abundance of some vOTUs remained very stable during ripening. They mainly corresponded to virulent *Lactococcus* phages, belonging to the *Skunavirus* genus (formerly 936 group). Among *Lactococcus* phages, this genus is by far the most frequently detected in the dairy industry (50). Given that *Lactococcus lactis* predominantly proliferates during milk acidification and maintains consistent concentration levels throughout ripening in this type of cheese (13), we theorize that the maintenance of *Skunavirus* is indicative of the stability of phage particles produced at the onset of cheese maturation. The remarkable stability of *Skunavirus* particles in comparison to other *Lactococcus* phage groups, especially the P335 group, has been reported earlier (51, 52). By contrast, we suggest that the relative decline observed in our experiment for other *Lactococcus* phages, specifically those affiliated with the P335 group, denotes their temporal instability. Nevertheless, a more comprehensive examination of this phenomenon warrants dedicated investigations.

Microbial communities of washed-rind cheese are generally dominated by environmental microorganisms detected in processing environments, the so-called "house" microbiota (53), which is specific to each production facility and provides a microbial signature distinguishing cheeses belonging to the same variety but manufactured by distinct producers (10). Recently, the analysis of several cheeses from Quebec revealed

that the dominant microorganisms remain stable from year to year, which could be linked to typical manufacturing practices and consistency in the use of starter and ripening cultures by cheesemakers (54). Here, we describe a similar observation for bacteriophages. We indeed identified a core virome composed of the most abundant vOTUs, consistently detected in three different years of production. Previous work on an undefined starter culture used for the production of a Swiss-type cheese propagated for decades in the same dairy environment revealed that phages and bacteria stably coexist over time in this system and suggests that this may contribute to the stable maintenance of the cheese starter culture over years (47). Phages and ripening bacteria present on the surface of natural smear-ripened cheese are contaminating the cheese production environment and are therefore likely to repeatedly contaminate cheese from one production cycle to another (34, 53). Nevertheless, further research is needed to elucidate the contributions of phages to both the long-term maintenance of the strain-level diversity of ripening bacteria and the occurrence of punctual microbial disequilibrium leading to organoleptic defects. These findings may be specific to the type of cheese under study, the strain-level diversity of the main bacteria, and the operational conditions used for cheese production.

In conclusion, the observed dynamics of the cheese virome throughout the ripening process, coupled with the persistence of the most dominant phages across production years, support the important role of bacteriophages in the cheese microbial ecosystem. The acquisition of new knowledge about phage ecology is now essential for a comprehensive understanding of microbial successions during milk fermentation. Moreover, this knowledge may offer cheesemakers novel avenues to refine and control their production processes.

## MATERIAL AND METHODS

### Cheese samples

#### Short-term study design

French smear-ripened cheeses, all from the same production batch, were collected directly from a cheese plant at five different stages during the ripening process (Fig. 1). Three different cheeses were sampled per stage and considered biological replicates. The stages, labeled W1 to W5, correspond to distinct washing steps. The initial three washes utilized a NaCl solution, while the final two employed a NaCl solution supplemented with increasing concentrations of alcoholic liquor (standard alcohol concentration of 40%). At each stage, three distinct cheeses were sampled and designated as biological replicates A, B, and C. Cheeses were immediately stored at 4°C after sampling and processed within 48 h.

#### Long-term study design

Ready-to-consume cheeses of the same type and brand as the previously described cheeses were purchased in a local supermarket in December 2017, November 2019, and February 2022, spanning >4 years of production (Fig. 1). Three different cheeses, with the same production date, were sampled each year and used as biological replicates. Cheeses were immediately stored at 4°C after sampling and processed within 48 h.

For both short-term and long-term studies, cheese samples were analyzed immediately after reception at the laboratory. Using sterile knives, the rind, approximately 2–3 mm thick, was carefully separated from the core. It was then blended and processed for microbial counts, viral DNA extraction for metavirome analysis, and microbial DNA extraction for extensive amplicon sequencing targeting the 16S rRNA gene, which will be subsequently referred to as 16S metabarcoding.

### Microbiological analysis

Bacteria were enumerated by plating serial dilutions ($10^{-1}$ to $10^{-7}$) of 1 g of cheese rind mixed in 9 mL of physiological water (9 g/L NaCl) on two different culture media as

described in our previous study (29). Brain heart infusion agar (BHI; Biokar Diagnostics) supplemented with 50 mg/L amphotericin (Sigma Aldrich, Saint Louis, MO, USA) was used to count total aerobic bacteria after 48 h of incubation at 28°C. Man, Rogosa, and Sharpe agar (MRS; Biokar Diagnostics, Allonne, France) supplemented with 50 mg/L amphotericin was used to count lactic acid bacteria after 48 h of incubation at 30°C under anaerobic conditions.

## Viral DNA extraction and metavirome analysis

Extraction of the viral fraction from the cheese rind was performed according to protocol P4 detailed in reference 29 including a filtration step and a chloroform treatment. DNA was extracted from the viral particles according to the protocol described in the same study and sent to Eurofins Genomics for high-throughput sequencing using the Illumina NovaSeq platform (2 × 150 bp paired-end reads, approximately 10 million reads per sample). The experimental procedures used for nucleic acids extraction and sequencing library preparation limited this study to the description of double-stranded DNA viruses, excluding single-stranded DNA and RNA viruses.

All the details about the tools, versions, and parameters used in the following pipeline are available in scripts deposited in the GitLab repository (https://forge-mia.inra.fr/eric.dugat-bony/cheese_virome). Briefly, raw reads were quality-filtered using Trimmomatic v0.39 (55). Then a single assembly was computed for the collection of triplicate reads from each sample with Spades v3.15.3 (56), using either the complete data set of trimmed reads available or after subsampling the data set to 1.5 million, 150,000, or 15,000 trimmed reads per sample. We noted that some abundant contigs were assembled into longer, nearly complete contigs, after subsampling. Among the 5,753 contigs, those >2 kb were selected and clustered following an approach adapted from reference 57. Succinctly, a pairwise alignment was first performed for all contigs using BLAT (58). Then, contigs with a self-alignment >110% of contig length, corresponding to chimeras, were removed. The remaining contigs were clustered at the species level (90% identity × coverage) and the longest contig within each cluster was selected as the representative sequence. The final contig data set consisted of 3,122 dereplicated contigs.

Among them, 332 were identified of viral origin (vOTUs) by using a combination of three detection tools: VIBRANT v1.2.1 (36), VirSorter2 v2.2.4 (59), and CheckV v0.8.1 (35). To be declared as viral, a contig had to meet at least one of the following criteria: declared "complete," "high," or "medium" quality by either VIBRANT or CheckV, declared "full" by VirSorter2. The bacterial host of the vOTUs was predicted using iPHoP (60). Finally, all sequences were compared by BLAT to an in-house database consisting of genome sequences from 32 common dairy phages (listed in Table S2) to identify potentially related phages with known taxonomy, verified host, and lifestyle (30% identity × coverage minimal cutoff). When appropriate, the host genus predicted by iPHoP was replaced by the genus of the bacterial host of the closest relative phage identified by the BLAT search. One vOTU, corresponding to the genome of the phage PhiX174 which is routinely used as control in Illumina sequencing runs to monitor sequencing quality, was discarded resulting in a final data set of 331 vOTUs.

To evaluate the relative abundance of each vOTU in each metavirome sample, trimmed reads were mapped against the vOTUs using bwa-mem2 v2.2.1 (61) and counted with Msamtools v1.0.0 profile with the options --multi=equal—unit --unit=ab –nolen (https://github.com/arumugamlab/msamtools). The output files from all samples were joined into an abundance table and processed using the R package phyloseq v1.38.0 (62). For each experimental data set (short-term and long-term studies), read counts were rarefied to the minimum depth observed in one individual sample. To facilitate comparison of abundance levels among vOTUs, read counts were divided by contig length resulting in an abundance table containing the mean coverage per vOTU in each sample. Subsequently, normalization of the table was conducted using the total sum scaling method (TSS) (63), which simply transforms the abundance table into a

relative abundance table through scaling by each sample's library size, thus enabling comparisons between samples. Assuming that one vOTU corresponds to one species (which is wrong each time several contigs belonging to the same phage genome are present), Bray-Curtis dissimilatory index, as computed by the distance function from the R package phyloseq, was used to compare viral communities between samples and the effect of different variables on their structure was assessed using permutational analysis of variance (PERMANOVA) as computed by the adonis2 function from the R package vegan v2.6–2.

Differential analysis was performed using the DESeq function implemented in the R package DESeq2 v1.38.0 (37) for which an official extension exists within the R package phyloseq (https://joey711.github.io/phyloseq-extensions/DESeq2.html). This method was shown to perform well in the detection of differentially abundant taxa in microbiome data (38). Because the DESeq function already includes a normalization step (by the median of ratios method), this analysis started from raw counts. The Wald test was used for comparing the abundance of all vOTUs between two sample groups and vOTUs were considered differentially abundant if the $p$ value was < 0.01 after correction for multiple testing using the Benjamini and Hochberg method, log2 fold change was >3 or <−3 and average raw counts was >1,500.

## Microbial DNA extraction and 16S metabarcoding profiles

Total DNA extraction from the cheese surface was performed as previously described (10). PCR amplification of the V3-V4 regions of the 16S rRNA gene was performed with the primers V3F (5′-ACGGRAGGCWGCAG-3′) and V4R (5′-TACCAGGGTATCTAATCCT -3′) carrying the Illumina 5′-CTTTCCCTACACGACGCTCTTCCGATCT-3′ and the 5′-GGAGT TCAGACGTGTGCTCTTCCGATCT-3′ tails, respectively. The reaction was performed using 10 ng of extracted DNA, 0.5 µM primer, 0.2 mM dNTP, and 2.5 U of the MTP Taq DNA polymerase (Sigma-Aldrich, USA). The amplification was carried out using the following program: 94°C for 60 s, 30 cycles at 94°C for 60 s, 65°C for 60 s, 72°C for 60 s, and a final elongation step at 72°C for 10 min. The resulting PCR products were sent to the @BRIDGe platform (INRAE, Jouy-en-Josas, France) for library preparation and sequencing. Briefly, amplicons were purified using magnetic beads CleanPCR (Clean NA, GC biotech B.V., The Netherlands), the concentration was measured using a Nanodrop spectrophotometer (Thermo Scientific, USA), and the amplicon quality was assessed on a Fragment Analyzer (AATI, USA) with the reagent kit ADNdb 910 (35–1,500 bp). Sample multiplexing was performed by adding tailor-made 6 bp unique indexes during the second PCR step which was performed on 50–200 ng of purified amplicons using the following program: 94°C for 10 min, 12 cycles at 94°C for 60 s, 65°C for 60 s, 72°C for 60 s, and a final elongation step at 72°C for 10 min. After purification and quantification (as described above), all libraries were pooled with equal amounts to generate an equivalent number of raw reads for each library. The DNA concentration of the pool was quantified on a Qubit Fluorometer (Thermofisher Scientific, USA) and adjusted to a final concentration between 5 and 20 nM for sequencing. The pool was denatured (NaOH 0.1N) and diluted to 7 pM. The PhiX Control v3 (Illumina, USA) was added to the pool at 15% of the final concentration, as described in the Illumina procedure, and the mixture was loaded onto the Illumina MiSeq cartridge according to the manufacturer's instructions using the MiSeq Reagent Kit v3 (2 × 250 bp paired-end reads).

Paired-end reads were analyzed using FROGS v3.2 (64), according to the standard operating procedure. Briefly, operational taxonomic units (OTUs) were built using Swarm with an aggregation distance of 1 and the --fastidious option (65), and each OTU that accounted for <0.005% of the total data set of sequences was discarded, as previously recommended (66). Lastly, OTU affiliation was checked using the EzBiocloud database v52018 (67). The abundance table was processed using the R package phyloseq v1.38.0 (62).

## ACKNOWLEDGMENTS

We are grateful to the INRAE MIGALE bioinformatics facility (doi: 10.15454/1.5572390655343293E12) for providing computing and storage resources.

T.P. is the recipient of a doctoral fellowship from the French Ministry of Higher Education, Research and Innovation (MESRI) and the MICA department of the French National Research Institute for Agriculture, Food and Environment (INRAE). For the metabarcoding analysis, this work has benefited from the facilities and expertise of @BRIDGe (Université Paris-Saclay, INRAE, AgroParisTech, GABI, Jouy-en-Josas, France).

## AUTHOR AFFILIATIONS

[1]Université Paris-Saclay, INRAE, AgroParisTech, UMR SayFood, Palaiseau, France
[2]Université Paris-Saclay, INRAE, AgroParisTech, Micalis Institute, Jouy-en-Josas, France

## PRESENT ADDRESS

Clarisse Figueroa, Université de Paris Cité, INSERM, IAME, UMR 1137, Paris, France

## AUTHOR ORCIDs

Quentin Lamy-Besnier http://orcid.org/0000-0002-7141-6340
Marie-Agnès Petit http://orcid.org/0000-0003-2242-2768
Eric Dugat-Bony http://orcid.org/0000-0002-5182-0063

## DATA AVAILABILITY

Raw sequencing data for cheese virome and bacterial microbiome were deposited at the Sequence Read Archive (SRA) of the NCBI (https://www.ncbi.nlm.nih.gov/sra/) as part of BioProjects PRJNA984302 (short-term study) and PRJNA984735 (long-term study).

## ADDITIONAL FILES

The following material is available online.

### Supplemental Material

**Supplemental figures (mSystems00201-24-s0001.docx).** Figures S1 to S3.
**Supplemental tables (mSystems00201-24-s0002.xlsx).** Tables S1 and S2.

### Open Peer Review

**PEER REVIEW HISTORY (review-history.pdf).** An accounting of the reviewer comments and feedback.

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
