## [Reviewer comments · mSystems]

Dynamics of the viral community on the cheese surface during maturation and persistence across production years

Paillet Thomas, Quentin Lamy-Besnier, Clarisse Figueroa, Marie-Agnès Petit, and Eric Dugat-Bony

Corresponding Author(s): Eric Dugat-Bony, Institut national de recherche pour l'agriculture l'alimentation et l'environnement

Review Timeline:

Submission Date:	February 9, 2024
Editorial Decision:	March 15, 2024
Revision Received:	April 5, 2024
Accepted:	May 6, 2024

Editor: Benjamin Wolfe

Reviewer(s): The reviewers have opted to remain anonymous.

Transaction Report:

DOI: <https://doi.org/10.1128/msystems.00201-24>

Re: mSystems00201-24 (Dynamics of the viral community on the cheese surface during maturation and persistence across production years)

Dear Dr. Eric Dugat-Bony:

Thank you for submitting a revised version of your manuscript. Reviewer #2 has provided some very important and helpful additional comments (see below) that should be addressed in a revised manuscript. I have also provided some comments and edits in an attached Word document. Please also address the comments and edits I made in your revised manuscript.

As Reviewer #2 notes, there are still some writing issues that should be smoothed out to improve the clarity of the manuscript. I tried to point out many of these in the attachment, but you should also carefully read the revised manuscript to make sure you have good paragraph structure and grammar.

Revision Guidelines

Sincerely,
Benjamin Wolfe
Editor
mSystems

Reviewer #2 (Comments for the Author):

The second round of reviews (reviewer 2) for "Dynamics of the viral community on the cheese surface during maturation and persistence across production years"

I acknowledge the extensive changes made to the manuscript. It seems like the points raised were either addressed or the reasoning explained. I therefore thank the authors. However, as the manuscript is largely re-written, I will not respond to the point-to-point responses but go through it again. Overall, the manuscript has clearly developed nicely. The combination of bacterial and viral analysis of the rind as well as studying the stability over short and long time scales is clearly of great interest.

I have three main points I want to mention before going into the details:

1. The manuscript is currently still lacking some grammatical and wording rigor.
2. I suggest to reconsider the definition of your long-time scale as you are not actually resampling the same cheese but rather looking for a similar trend in different cheeses over different production years and cheese factories.
3. It is sometimes not clear why you are doing certain analysis or using certain wording (see mentioned points below)

For reference, I am referring to the line numbers in the Marked-up manuscript.

Abstract:

overall good.

L17: I wonder if you mean co-exist. As this to mean suggests the same phage to persist. Or do you mean that bacteriophages are a common and substantial part of the smear cheese microbiome. Maybe clarify if you mean the latter.

L20: detail but I am wondering if it is the ripening period or just cheese ripening.

L28: " a core of abundant vOTUs was systematically detected" is a bit of a wordy formulation, maybe "certain phages were observed throughout."

L29: Not sure what the "main phages" means. Maybe you go into this latter. Maybe "keystone" ? But then you have to clearly show that they are the "puppet masters".

Introduction:

L53: "peculiar" is not a very clear word. Do you mean distinct or unique or diverse or ...

L63: You mention that the bacteria come from the starter culture but then you do not mention where the other taxons come from. I think this could be important as the phages likely have a very similar origin. It is an open system not a sterile environment.

L74: You mention three reasons for microbial succession. Afterwards you explain what has been studied but you do not make your crucial link why phages should be studied as key ecological drivers. Maybe it is clear for you but just don't leave it unmentioned for the reader.

L90: You mention the "ecology of the phage". As this is a central part of you question and argumentation I would opt to define exactly what you mean with this. Do you mean the role of the phage in maintaining diversity or community composition overall?

L101: I am not sure if "on the short-term" is the right formulation maybe rather say,

Results1

L112: I suggest to split the sentence in two parts at the "and".

L114: This seems very strongly methods heavy. I would opt to just leave it for the methods or make it shorter.

L123: wow nice! That is a lot of mapping. Can you be sure the rest is really bacterial or other phages which did not assemble properly (e.g. <2kb contigs).

L129: maybe add on which taxonomic level these were predicted (genus-level).

Results 2:

L151: It is hard to understand the meaning of the diversity from only a shannon diversity number. You need to compare the diversity between timepoints or treatments...

L155: what does sparcity of 29% mean? 29% of phages where sample specific?

L160: Relative abundance of 0.005 %? are you able to faithfully predict the presence of these phages? Did the entire contig have mapping at what coverage?

L162: Sentence is grammatically not correct.

Fig 3C: cluster should be labeled more clearly in figure.

Results 3:

L221-L228: this is very complicated formulation. Do I understand correctly, your phage abundance and host abundance correlates positively? What does that ecologically mean? Are these temperate phages so if the host is high the phage is high. The phage does not seem to have a negative impact on the host. This is likely because you are looking at the genus level, right? Dynamics are probably more pronounced at the species or strain level.

Results 4:

I am not sure your can frame it as long-term stability. You are looking at different cheese varieties and different producers. It is not like you follow the same cheeses. Maybe think of a better terminology. Maybe Generality or reproducibility of the phage diversity?

L272: Not sure you can talk about structural variations here. These are compositional changes.

Discussion:

L306: Do you have any indications of the viral particle count?

Editor comments:

L18: Careful here and elsewhere when you say you study the ecological significance or role of the phages in this system. You don't really do that. You just characterize their diversity and associations with bacteria. To really test their ecological significance, you would need to do manipulations/experiments.

Answer: we agree with the editor and rephrased the sentence accordingly.

L34: A lot of this repeats what you already say in the abstract. Please streamline the Importance section to no be repetitive.

Answer: we revised the importance section accordingly.

L52: these are subjective terms and these communities are actually not that diverse relative to many other microbiomes. Use more precise terms here or cut these words

Answer: we revised the sentence according to this comment and the one from reviewer 2.

L96: Can you elaborate a bit more here? What specifically is not know about their ecology? Help the reader understand the research gap you are trying to fill.

Answer: according to the comment, we emphasized the fact that knowledge about temporal variations of the phage communities in cheese is lacking.

L105: Can you add a one sentence preview of findings here to help better close off the Introduction?

Answer: we followed the suggestion of the editor and added a sentence to close off the introduction.

Reviewer #2 (Comments for the Author):

The second round of reviews (reviewer 2) for "Dynamics of the viral community on the cheese surface during maturation and persistence across production years"

I acknowledge the extensive changes made to the manuscript. It seems like the points raised were either addressed or the reasoning explained. I therefore thank the authors. However, as the manuscript is largely re-written, I will not respond to the point-to-point responses but go through it again. Overall, the manuscript has clearly developed nicely. The combination of bacterial and viral analysis of the rind as well as studying the stability over short and long time scales is clearly of great interest.

Answer: we would like to thank again Reviewer 2 for the useful comments that contributed to improve the quality of the manuscript.

I have three main points I want to mention before going into the details:

1. The manuscript is currently still lacking some grammatical and wording rigor.

Answer: we carefully revised the manuscript.

2. I suggest to reconsider the definition of your long-time scale as you are not actually resampling the same cheese but rather looking for a similar trend in different cheeses over different production years and cheese factories.

Answer: Yes, indeed, this is what has been done. All cheeses analyzed were sourced from the same factory (same brand, same type of cheese). Although the experimental design did not involve resampling the same cheese in 2017, 2019, and 2022 (the usual ripening time on this type of cheese is about 4 weeks), our samples represented successive production batches from the same production line, thus equivalent to three production batches of the same type of cheese occurring at ~ two-year intervals. In our opinion, this design enables to evaluate the temporal variation of the phage community in this ecosystem (the rind of this specific type of cheese) over the years of production.

3. It is sometimes not clear why you are doing certain analysis or using certain wording (see mentioned points below)

Answer: we tried to revise the manuscript according to the comments.

For reference, I am referring to the line numbers in the Marked-up manuscript.

Abstract:

overall good.

L17: I wonder if you mean co-exist. As this to mean suggests the same phage to persist. Or do you mean that bacteriophages are a common and substantial part of the smear cheese microbiome. Maybe clarify if you mean the latter.

Answer: we rephrased this sentence according to this comment and the one from the editor.

L20: detail but I am wondering if it is the ripening period or just cheese ripening.

Answer: we replaced ripening period by cheese ripening here and also in the discussion (L331).

L28: " a core of abundant vOTUs was systematically detected" is a bit of a wordy formulation, maybe "certain phages were observed throughout."

Answer: we modified the sentence accordingly.

L29: Not sure what the "main phages" means. Maybe you go into this latter. Maybe "keystone" ? But then you have to clearly show that they are the "puppet masters".

Answer: it means the dominant ones (as revealed in Results 4), so we replaced “main” by “dominant” in the sentence for better clarity.

Introduction:

L53: "peculiar" is not a very clear word. Do you mean distinct or unique or diverse or ...

Answer: we revised the sentence according to this comment and the one from the editor.

L63: You mention that the bacteria come from the starter culture but then you do not mention where the other taxons come from. I think this could be important as the phages likely have a very similar origin. It is an open system not a sterile environment.

Answer: thank you for the comment. The other taxa (yeasts and ripening bacteria) in smear-ripened cheeses may originate from different sources such as raw milk, ripening cultures or the manufacturing environment. We added this information.

L74: You mention three reasons for microbial succession. Afterwards you explain what has been studied but you do not make your crucial link why phages should be studied as key ecological drivers. Maybe it is clear for you but just don't leave it unmentioned for the reader.

Answer: we added some precision at the end of the paragraph. We thought that phages can be responsible for the elimination of key bacterial populations and therefore disturb the overall succession of microbial communities in cheese.

L90: You mention the "ecology of the phage". As this is a central part of you question and argumentation I would opt to define exactly what you mean with this. Do you mean the role of the phage in maintaining diversity or community composition overall?

Answer: according to this comment and the one from the editor, we added a sentence to explain more precisely what gap into the bacteriophage ecology we are trying to fill.

L101: I am not sure if "on the short-term" is the right formulation maybe rather say,

Answer: this comment is not complete. But we modified the way we introduced our experimental design at the end of the introduction.

Results1

L112: I suggest to split the sentence in two parts at the "and".

Answer: we did the modification.

L114: This seems very strongly methods heavy. I would opt to just leave it for the methods or make it shorter.

Answer: we moved this sentence to the methods section.

L123: wow nice! That is a lot of mapping. Can you be sure the rest is really bacterial or other phages which did not assemble properly (e.g. <2kb contigs).

Answer: thanks for this comments, that's true. We are not sure the rest originates from bacterial contamination. We re-worded the sentence accordingly.

L129: maybe add on which taxonomic level these were predicted (genus-level).

Answer: we added this information.

Results 2:

L151: It is hard to understand the meaning of the diversity from only a shannon diversity number. You need to compare the diversity between timepoints or treatments...

Answer: we modified the first sentence of the paragraph to make it clearer. We actually compared the diversity (estimated by the Shannon index) between timepoints and performed the appropriate statistical test. See L160-161: “the decrease was not statistically significant, suggesting that there was no major alteration of the viral diversity related to the washing operations ($p > 0.05$, Kruskal-Wallis test; Fig. 3A).”

L155: what does sparsity of 29% mean? 29% of phages where sample specific?

Answer: the sparsity refers to the percentage of 0 values in the abundance matrix. In our case the percentage is quite low (29%) which indicates that most of the vOTUs have positive counts (> 1 read) in most of our samples. We added the definition of sparsity percentage in the corresponding sentence.

L160: Relative abundance of 0.005 %? are you able to faithfully predict the presence of these phages? Did the entire contig have mapping at what coverage?

Answer: we did not perform this type of coverage analysis to ascertain the presence of a given contig in a given sample. Our 0.005% threshold, which is applied on the sum of all relative abundances in a given subset (short-term study or long-term study), had the purpose to eliminate from each of the two subsets the contigs which were only present in one, but not the two of them. It basically helped for the visualisation of the heatmap. A rough calculation indicates that this average 0.005% relative abundance represents at least 1285 reads for the selected contigs. Given 1 read is 150 bases long, we can estimate a minimal coverage of approx. 96 for a selected vOTU of 2 kb, 38 for a vOTU of 5 kb, and 19 for a vOTU of 10 kb. We want to stress that this cut-off was not applied before any statistical test (alpha, beta diversity etc...). The purpose was just to reduce the number of contigs to print on the heatmap (for easier visualization).

L162: Sentence is grammatically not correct.

Answer: we corrected the sentence.

Fig 3C: cluster should be labeled more clearly in figure.

Answer: we increased the size of the cluster labels in figure 3C.

Results 3:

L221-L228: this is very complicated formulation. Do I understand correctly, your phage abundance and host abundance correlates positively? What does that ecologically mean? Are these temperate phages so if the host is high the phage is high. The phage does not seem to have a negative impact on the host. This is likely because you are looking at the genus level, right? Dynamics are probably more pronounced at the species or strain level.

Answer: we rephrased to make this part clearer. Reviewer 2 is right, the dynamics is likely to occur at lower taxonomic levels. We didn't conclude (and speculate) about the ecological meaning of this result since we didn't work at the appropriate taxonomic level. There is a body of literature about undefined starter cultures and swiss-type cheeses showing that while phages effectively regulates bacterial populations (strains-level), multi-strains communities are generally resilient and remain stable in terms of species or functional groups after phage infections. This is mentioned in the discussion section (L319-330).

Results 4:

I am not sure your can frame it as long-term stability. You are looking at different cheese varieties and different producers. It is not like you follow the same cheeses. Maybe think of a better terminology. Maybe Generality or reproducibility of the phage diversity?

Answer: In fact, we are looking at the same cheese variety, and the same producer. All the cheeses used for the long-term study were exactly of the same type and brand (meaning from the same producer) but were collected at different years of production. They are therefore directly comparable and, based on the high percentage of shared vOTUs across production years (56.6% of the total or 89.9% if we consider only the most abundant ones), we strongly believe that the term "stability" is appropriate regarding the composition of the virome (presence/absence).

L272: Not sure you can talk about structural variations here. These are compositional changes.

Answer: we modified the sentence accordingly.

Discussion:

L306: Do you have any indications of the viral particle count?

Answer: unfortunately we did not perform viral particles count on the studied samples. In a previous work (<https://doi.org/10.1016/j.fm.2019.103278>), we estimated the number of nanoparticles in cheese rind using an interferometric light microscope and found values around 10^{10} per gram. However, this value may be overestimated since the method is not selective against viruses and may also count other nanoparticles potentially present in cheese (e.g. membrane vesicles).

Re: mSystems00201-24R1 (Dynamics of the viral community on the cheese surface during maturation and persistence across production years)

Dear Dr. Eric Dugat-Bony:

I am pleased to inform you that your manuscript has been accepted, and I am forwarding it to the ASM production staff for publication.

Please note that this email includes an attachment of the main text of your manuscript with some minor text changes I suggest you make during the proofing stage. These are minor edits to improve clarity or grammar.

Your paper will first be checked to make sure all elements meet the technical requirements. ASM staff will contact you if anything needs to be revised before copyediting and production can begin. Otherwise, you will be notified when your proofs are ready to be viewed.

Cover Image Submissions: If you would like to submit a potential Cover Image, please email a file and a short legend to mssystems@asmusa.org. Please note that we can only consider images that (i) the authors created or own and (ii) have not been previously published. By submitting, you agree that the image can be used under the same terms as the published article. Image File requirements: TIF/EPS, 7.5 inches wide by 8.25 inches tall (at least 2,250 pixels wide by 2,475 pixels tall), minimum 300 dpi resolution (600 dpi preferred), RGB, and no figure elements, e.g., arrows or panel labels. The legend should be a short description of the image, 1-2 sentences recommended.

We recognize that the video files can become quite large, so to avoid quality loss ASM suggests sending the video file via <https://www.wetransfer.com/>. When you have a final version of the video and the still ready to share, please send it to mSystems staff at mssystems@asmusa.org.

Sincerely,
Benjamin Wolfe
Senior Editor
mSystems